# NU HLT at CMCL 2022 Shared Task:
# Multilingual and Crosslingual Prediction of Human Reading Behavior in Universal Language Space

**Joseph Marvin Imperial**

Human Language Technology Lab (NU HLT)

National University

Manila, Philippines

jrimperial@national-u.edu.ph

## Abstract

In this paper, we present a unified model that works for both multilingual and crosslingual prediction of reading times of words in various languages. The secret behind the success of this model is in the preprocessing step where all words are transformed to their universal language representation via the International Phonetic Alphabet (IPA). To the best of our knowledge, this is the first study to favorably exploit this phonological property of language for the two tasks. Various feature types were extracted covering basic frequencies, n-grams, information theoretic, and psycholinguistically-motivated predictors for model training. A finetuned Random Forest model obtained best performance for both tasks with 3.8031 and 3.9065 MAE scores for mean first fixation duration (FFDAvg) and mean total reading time (TRTAvg) respectively[1].

## 1 Introduction

Eye movement data has been one of the most used and most important resource that has pushed various interdisciplinary fields such as development studies, literacy, computer vision, and natural language processing research into greater heights. In a technical point of view, correctly determining theoretically grounded and cognitively plausible predictors of eye movement will allow opportunities to make computational systems leveraging on these properties to be more human-like (Sood et al., 2020).

Common human reading prediction works make use of the standard Latin alphabet as it is internationally used. However, investigating eye movement and reading patterns in other non-Anglocentric writing systems such as Chinese and Bengali is as equally as important (Share, 2008; Liversedge et al., 2016). Fortunately, there is a growing number of previous works exploring multilinguality in eye tracking prediction both in data collection and novel prediction approaches. The study of Liversedge et al. (2016) was the first to explore potential crosslinguality of Chinese, English and Finnish which differ in aspects of visual density, spacing, and orthography to name a few. The results of the study favorably support possible *universality of representation* in reading. In the same vein, Hollenstein et al. (2021) was the first to try use of large finetuned multilingual language models like BERT (Devlin et al., 2019) and XLM (Conneau and Lample, 2019) in a crosslingual setting to predict eye tracking features across English, Dutch, German, and Russian. Data-wise, the published works of Siegelman et al. (2022) for MECO, Pynte and Kennedy (2006) for the Dundee corpus, and Cop et al. (2017) for GECO have made significant impact in the field where they covered curation and collection of eye-tracking corpus for other languages in addition to English.

## 2 Task Definition and Data

The CMCL 2022 Shared Task (Hollenstein et al., 2022)[2] describes two challenges: predicting eye-tracking features in a **multilingual** and **crosslingual setup**. The eye movement dataset for this Shared Task contains sentences written in six languages: Mandarin Chinese (Pan et al., 2021), Hindi (Husain et al., 2015), Russian (Laurinavichyute et al., 2019), English (Luke and Christianson, 2018; Hollenstein et al., 2018, 2020), Dutch (Cop et al., 2017), and German (Jäger et al., 2021). The mean first fixation duration (FFDAvg) and mean total reading time (TRTAvg) as well as their corresponding standard deviations (FFDStd and TRTStd) are the four main eye-tracking features that need to be predicted by the participants through proposed computational means. For the multilingual task,

---

[1] https://github.com/imperialite/cmcl2022-unified-eye-tracking-ipa

[2] https://cmclorg.github.io/shared_task

the training, validation, and testing datasets conform to the identified six languages. While for the crosslingual task, a surprise language (Danish) is provided as the test dataset.

## 3 Eye-Tracking Prediction in Universal Language Space

The proposed solution in this work is inspired by both classical and recent previous works in speech recognition systems (Schultz and Waibel, 1998, 2001; Dalmia et al., 2019) with multilingual and crosslingual capabilities through the transformation of words or similar sounding units in one global shared space using the International Phonetic Alphabet (IPA). This functionality allows models to generalize and adapt parameters to new languages while maintaining a stable vocabulary size for character representation. By definition, the IPA contains 107 characters for consonants and vowels, 31 for diacritics for modifying said consonants and vowels, and 17 signs to emphasize suprasegmental properties of phonemes such as stress and intonation (Association et al., 1999).

Figure 1 describes the unified methodology used for tackling both the multilinguality and crosslinguality challenge of the Shared Task. The backbone of this proposed solution lies with the phonetic transcription preprocessing step to convert the raw terms from the data written in Mandarin Chinese, Hindi, Russian, English, Dutch, and German to their IPA form. We used Epitran by Mortensen et al. (2018) for this process. The surprise language for the crosslingual task, Danish, is not currently supported by Epitran. We instead resorted to use Automatic Phonetic Transcriber[3], a paid transcription service that caters the Danish language. The transcription cost of the Danish test data is €15.

### 3.1 Feature Extraction

After obtaining the phonetic transcriptions, a total of fourteen features based on various types were extracted spanning general frequencies, n-grams, based on information theory, and based on motivations from psycholinguistics.

**Frequency and Length Features**. The simplest features are frequency and length-based predictors. Studies have shown that the length of words correlate with fixation duration as long words would obviously take time to read (Rayner, 1977;

---

[3] http://tom.brondsted.dk/text2phoneme/

---

Hollenstein and Beinborn, 2021). For this study, we extracted the (a) word length (word_len), (b) IPA length (ipa_len), (c) IPA vowels count per term (ipa_count), and (d) normalized IPA vowel count per term over length (ipa_norm).

**N-Gram Features**. Language model-based features is a classic in eye-tracking prediction research as they capture word probabilities through frequency. We extracted raw count of unique n-grams per word (bigram_count, trigram_count), raw count of total n-grams per term (bigram_sum, trigram_sum), and normalized counts over word length (bigram_norm, trigram_norm) for character bigrams and trigrams in IPA form guided by the general formula for n-gram modelling below:

$$P(w_n \mid w_{n-N+1}^{n-1}) = \frac{C(w_{n-N+1}^{n-1}w_n)}{C(w_{n-N+1}^{n-1})} \quad (1)$$

**Psycholinguistially-Motivated Features**. Features with theoretical grounding are more practical to use when invetigating phenomena in human reading. In line with this, we extracted two psycholinguistically-motivated features: **imageability** and **concreteness**. When reading, humans tend to visualize words and scenarios as they are formed in context. This measure of ease of how words or phrases can easily be visualized in the min from a verbal material is quantified as imageability (Lynch, 1964; Richardson, 1976). On the other hand, concreteness is a measure of lexical organization where words are easily perceived by the senses. In the example of Schwanenflugel et al. (1988), words such as *chair* or *computer* are better understood than abstract words like *freedom*. Words with high concreteness scores are better recalled from the mental lexicon than abstract words as they have better representation in the imaginal system (Altarriba et al., 1999). We use these two features as we posit that the visualization and retrieval process of imageability and concreteness respectively can contribute to the reading time in milliseconds.

For this task, we used the crosslingual word embedding-based approximation for all the seven languages present in the dataset from the the work of Ljubešić et al. (2018).

**Information Theoretic Features**. Features inspired by information theory such as the concept

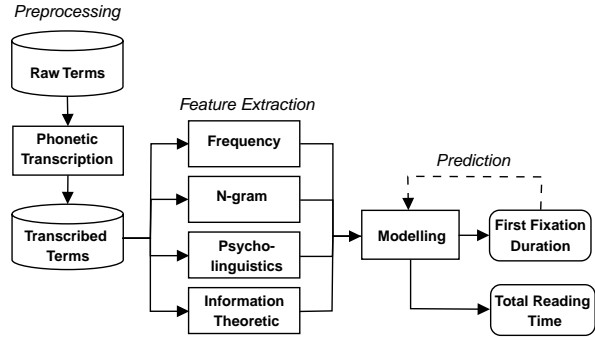

Figure 1: The proposed **unified** approach to multilingual and crosslingual human reading pattern prediction in universal language space via IPA.

of surprisal have thoroughly used in human reading pattern prediction (Hale, 2001; Levy, 2008; Demberg and Keller, 2008, 2009; Goodkind and Bicknell, 2018). Surprisal describes that processing time of a word to be read is proportional to its negative log based on a probability given by context as shown below:

$$\text{surprisal}(w_i) = -\log_2 P(w_i \mid w_1...w_{i-1}) \quad (2)$$

Thus, if a word is more likely to occur in its context, it is read more quickly (Shannon, 1948). For this task, since words are converted to a universal language space, the correct terminology in this case is bits per phoneme or **phonotactic complexity** as coined by Pimentel et al. (2020).

While surprisal quantifies the word's predictability or processing cost during reading, we also obtain the **entropy** $H$ of each word $x$ from the corpus. The entropy quantifies the expected value of information from an event as shown in the formula below:

$$H(X) = -\sum_{i=1}^{n} (\frac{count_i}{N}) \log_2 (\frac{count_i}{N}) \quad (3)$$

where $count_i$ is the count of character $n_i$ and each word $N$ consists of $n$ characters. With this measure, a higher entropy score entails higher uncertainty for a word, thus, leading to increased reading time at the millisecond level.

### 3.2 Model Training Setup

We used four machine learning algorithms via WEKA (Witten and Frank, 2002) for modelling the features with FFDAvg and TRTAvg: linear regression (**LinReg**), multilayer perceptron (**MLP**), random forest (**RF**), and k-Nearest Neighbors (**kNN**).

We only used the finetuned RF model for the prediction of FFDAvg and TRTAvg. Meanwhile, FFDStd and TRTStd are obtained by using the top models of all the four algorithms, re-running them to get FFDAvg and TRTAvg, and calculating the standard deviation. For TRTAvg, we added the predicted FFDAvg from the best model as an additional feature as we posit that the first fixation duration is a contributor to the overall reading time.

## 4 Results

Table 1 describes the main results of the experiments for predicting FFDAvg and TRTAvg using multiple finetuned supervised techniques evaluated through mean absolute error (MAE) and root mean squared error (RMSE). As mentioned previously, since the methodology used in this study cuts across multilingual and crosslingual tasks, the results reported in this applied are applicable to both. From the Table, the RF models outperformed the other three models in predicting FFDAVg and TRTAvg using 100% and 75% random selected features respectively and across 100 iterations. The RF model's effectivity can be attributed to its structure of multiple decision trees which normalize overfitting (Ho, 1995). Following RF in performance is kNN using Euclidean distance observing the same pattern as RF with different hyperparameter values such as 5 and 20 for the nearest neighbor for predicting FFDAvg and TRTAvg. On the other hand, both LinReg and MLP have no improvements regardless of hyperparameter values. For LinReg, using an M5 feature selection only provides extremely minor improvement in performances for FFDAvg and TRTAvg prediction. For MLP, using

| Model | FFDAvg | | TRTAvg | |
|---|---|---|---|---|
| | MAE | RMSE | MAE | RMSE |
| **LinReg (k=10, M5)*†** | **5.2361** | **6.7267** | **4.3419** | **7.0546** |
| LinReg (k=10, greedy) | 5.2361 | 6.7267 | 4.3420 | 7.0545 |
| LinReg (k=10, none) | 5.2363 | 6.7274 | 4.3429 | 7.0594 |
| **MLP (k=10, lr=0.005, m=0.2)*†** | **4.9898** | **6.4169** | **4.1744** | **6.2140** |
| MLP (k=10, lr=0.5, m=0.2) | 6.7916 | 8.3791 | 4.8475 | 7.0840 |
| MLP (k=10, lr=0.005, m=0.002) | 5.0018 | 6.4299 | 4.1862 | 6.2177 |
| MLP (k=10, lr=0.5, m=0.002) | 6.4447 | 8.0110 | 4.9528 | 6.9668 |
| MLP (k=10, lr=0.0005, m=0.0002) | 5.5024 | 7.0474 | 4.2956 | 6.3823 |
| **RF (k=10, iters = 100)*** | **3.8031** | **5.2750** | 3.9600 | 5.8446 |
| RF (k=10, iters = 100, 50% feats) | 3.8045 | 5.2766 | 3.9094 | 5.8015 |
| RF (k=10, iters = 100, 75% feats†) | 3.8056 | 5.2762 | **3.9065** | **5.8006** |
| **kNN (k=10, nn=5, dist=euc)*** | **4.3335** | **5.9651** | 4.2953 | 6.3741 |
| kNN (k=10, nn=10, dist=euc) | 4.4263 | 6.0133 | 4.2053 | 6.2436 |
| kNN (k=10, nn=20, dist=euc)† | 4.5646 | 6.1284 | **4.1793** | **6.2432** |

Table 1: Results of predicting mean first fixation duration (FFDAvg) and mean total reading time (TRTAvg) using hyperparameter-tuned traditional supervised models over cross-fold validation of $k$=10. The tuned Random Forest (RF) model achieved the best performance which was used for both tasks of multilingual and crosslingual prediction. Top performing models from the four algorithm class were used for predicting the held-out test data to get the standard deviation of FFDAvg (*) and TRTAvg (†).

| FFDAvg | | TRTAvg | |
|---|---|---|---|
| bigram_norm | -0.1751 | FFDAvg | 0.8068 |
| trigram_norm | -0.1393 | bigram_count | 0.2219 |
| word_len | -0.1334 | trigram_count | 0.2156 |
| bigram_sum | -0.1304 | phonetic_comp | -0.2107 |
| trigram_sum | -0.1101 | ipa_ent | 0.1925 |
| imageability | 0.1101 | ipa_len | 0.1921 |
| concreteness | 0.1044 | trigram_norm | -0.1886 |

Table 2: Top 7 predictors for FFDAvg and TRTAvg with the highest absolute correlation coefficients.

default values in WEKA for momentum and learning rate obtained the best performance similarly for for FFDAvg and TRTAvg prediction.

### 4.1 Feature Importance

Viewing the results in a correlation analysis perspective, Table 2 shows the top 50% of the predictors, total 7, which are significantly correlated with FFDAvg and TRTAvg respectively. Only one predictor is common for both values, the normalized trigrams in IPA space which is fairly high in FFDAvg along with normalized bigrams than in TRTAvg. This may hint that normalized n-gram features may be plausible features of eye movement only for first passes over the word and not with the total accumulated time of fixations. Likewise, the psycholinguistically-motivated features, imageability and concreteness, were only seen in the FFDAvg section as well proving their potential plausibility for the same observation. All the

length-based features such as word, IPA, bigram, and trigram-based counts were considered as top predictors for FFDAvg and TRTAvg. This unsurprisingly supports the results from the classical work of Rayner (1977) on correlation of lengths with fixations. Lastly, the strong correlation of first fixation duration with the total reading time with a score of $r = 0.8068$ proves the theoretical grounding of the proposed methodology as stated in Figure 1 albeit in post-hoc.

## 5 Conclusion

Precise eye movement datasets in multiple languages are considered one of the most important contributions that benefit various interdisciplinary fields such as psycholinguistics, developmental studies, behavioral studies, computer vision, and natural language processing. In this paper, we present a novel method of transforming multilingual eye-tracking data (English, Mandarin, Hindi, Russian, German, Dutch, and Danish) to their IPA equivalent, enforcing a single vocabulary space which allows competitive results for both multilingual and crosslingual tasks in a regression analysis setup. Future directions of this paper can explore more cognitively and theoretically plausible features that can be extracted as well as deeper interpretation studies of the predictive models trained.

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
