# OpenReview forum: "NU HLT at CMCL 2022 Shared Task: Multilingual and Crosslingual Prediction of Human Reading Behavior in Universal Language Space"
_aclweb.org/ACL/2022/Workshop/CMCL_Shared_Task — CMCL Shared Task_

### Official Review · Reviewer_MaeN · 2022-03-19
**Good descriptions of a working system**

**Rating:** 7
**Confidence:** 4

**Review:**

### Summary
This paper describes the NU-HLT system in CMCL 2022 shared task. This system is inspired by both classical and recent previous works in speech recognition systems. First, the raw terms are transformed into a global shared space using IPA. Then, some features, including the frequency, length, N-gram, and information-theoretic features are extracted. Specifically, there are two psycholinguistically-motivated features: imageability and concreteness. Four ML algorithms from WEKA (LinReg, MLP, RF, kNN) are used to train to predict the FFDAvg and TRTAvg scores. Additionally, top predictor features with their correlation coefficients are identified.

### Reasons to accept
- The description of the system is clear. The hyperparameters including learning rate and the number of iterations are reported, allowing easy reproduction of the results.
- The idea of transforming into IPA turns out to be novel and effective for predicting the FFDAvg and TRTAvg tasks.

### Reasons to reject
I see no serious issues with this paper.

### Comments
- The reporting of some results could be improved. E.g., in Table 1, there are many mentions of `k`, `m`. What do they mean?
- Table 2: highest correlation coefficients. Do you mean "highest absolute correlation coefficients"? Some numbers are negative there.
- Probably an additional feature selection procedure can be useful for further improving the predicting performance.
- Reference: Some entries have urls, but the remaining do not. Recommend adding (or removing) urls to all entries to keep the style consistent.

---

### Official Review · Reviewer_idfb · 2022-03-21
**Very well described details of the working system**

**Rating:** 7
**Confidence:** 3

**Review:**

## Summary

This paper describes the system of NU-HLT for the CMCL 2022 shared task. The system, inspired by previous works in speech recognition, uses a novel preprocessing step involving the transformation of words to a global vector space using IPA. Next fourteen features were then extracted from the transcriptions. The features included features like length and frequency of words and language-model based features in the form of different n-gram based statistics. Psychologically-motivated features in the form of imageability and concreteness were also extracted. Finally, information theoretic features in the form of surprisal was extracted for the systems. Using WEKA, four ML algorithms ( Linear regression, MLP, Random Forest and k-NN) were used to train the systems for predicting FFDAvg and TRTAvg features. Additionally, top 50% of the predictors are identified using their correlation with FFDAvg and TRTAvg.

## Reasons to accept

* The system description is very clear, concise and informative. All the details about the features, experiments and hyperparameters are mentioned.
* The idea of transforming the raw words into IPA is novel.

---

### Decision · Program_Chairs · 2022-03-28

Accept